# Data-Driven Design as a Vehicle for BIM and Sustainability Education

**John Benner and J. J. McArthur *** 

Department of Architectural Science, Ryerson University, Toronto, ON M5B 2K3, Canada;
john.benner@ryerson.ca
* Correspondence: jjmcarthur@ryerson.ca; Tel.: +1-416-979-5000

**Abstract:** The development of BIM pedagogical strategies within the Architecture, Engineering, and Construction disciplines is a topic of significant research. Several approaches and theoretical lenses, such as Project-Based Learning, constructivist pedagogy, experiential learning, and Bloom's Taxonomy have been applied to guide pedagogical education. This paper presents the development and evaluation of an approach integrating these four perspectives that was developed within an Architectural Science undergraduate program. A data-driven design project was incorporated into the curriculum to give students opportunities to engage with BIM-based simulation (cost and energy) to guide their design studio project development. The pedagogical approach is discussed, along with refinements to this project based on early implementation. Four years of data are analyzed, consisting of 1325 design iterations and student feedback on the project. A critical evaluation of the project determined that it was highly effective to engage students at an advanced level - level 4 (Analyze) of Bloom's Taxonomy was consistently achieved (over 96% of students) and two thirds of students also engaged meaningfully at Level 5 (Evaluate; 67%) and/or 6 (Create; 8%)—while developing a high degree of competence in the use of BIM.

**Keywords:** BIM education; project-based learning; constructivism; experiential learning; Bloom's taxonomy; data-driven design; sustainable design

## 1. Introduction

There is an increasing interest within the practice of architecture in "data-driven design", referring to the integration of simulation into the design process. Concurrently, there is active discourse on the best means to integrate BIM into architecture and engineering curricula at the university level, balancing a need to equip students with industrially-relevant skills while avoiding the devolution into the simple teaching of a software.

A breadth of approaches have been developed to integrate BIM into the architectural curriculum, including studio or design courses [1,2] construction applications [3], and capstone projects [4]. Abdirad and Dossick [5] present an excellent summary of BIM pedagogical approaches through March 2015 in their systematic review of BIM pedagogical practice, providing a valuable reference for educators. Poirier et al. [6] provide further insight obtained through a workshop on BIM education, bringing together Canadian, US, UK, and Australian perspectives and case studies. Several papers further address BIM as a teaching methodology for sustainable design [7] and/or construction management, including cost control [8].

This paper engages with BIM as a means to integrate data-driven design into the architecture curriculum and presents the ongoing development of a project-based learning exercise developed to guide students to gain advanced BIM skills in their third year of undergraduate study in Architectural Science. This is part of a BIM integration curriculum developed in 2015 to help students transcend the

learning of BIM as software, instead focusing the discussion on the rationale for the application of BIM within architecture: when and why it should or should not be used, how best to manage BIM on a complex project, and how such decisions impact the design process. A project was designed to introduce students to the use of BIM-based building performance simulation and cost analysis as an iterative design tool and integrated into the curriculum to both teach life-cycle costing concepts as well as provide insight on conceptual massing development in the concurrent studio course. As the architectural profession increasingly moves towards in-house simulation and data-driven design, there is significant interest in developing both the necessary skills and an understanding of the limitations of such practices within the architecture student body. This paper addresses both these student outcomes - through a critical evaluation of the student results - as well as the success of the learning process from a pedagogical perspective, and contributes to insights on how BIM and data-driven design can be taught conjointly at an undergraduate level. Despite the significant body of literature on BIM teaching and the application of simulation in architecture, there is a paucity of studies showing how these can be effectively combined within an undergraduate context. Given the urgent need to reduce the energy consumption of buildings and the known benefit of early stage energy modeling to inform massing and orientation, recently codified in ASHRAE Standard 209-2018 "Energy Simulation Aided Design for Buildings except Low Rise Residential Buildings" [9], this project also addresses a contemporary issue in sustainable building design.

## 2. Pedagogical Approaches to BIM Education

Within the academic literature, four pedagogical theories are frequently applied to BIM education: project-based learning (PBL), constructivism, experiential learning, and Bloom's taxonomy [10–19]. Each of these theories consider the active role of the learner and the acquisition of knowledge through the completion of a task with real-world applications. A brief description of each theory and pertinent examples of their application to BIM education follow.

### 2.1. Project-Based Learning

Project-based learning is defined as: "a comprehensive approach to classroom teaching and learning that is designed to engage students in investigation of authentic problems" [20]. A broad review of research in PBL was conducted by Thomas [10] and provides insight on the extent of this approach in education. This approach has been widely used for BIM education, particularly within the architecture and construction management context, either as individual or group work [6]. PBL is widely adopted and has been found to be more popular than traditional learning with both students and instructors [10]. There has been some adoption of PBL documented in the academic research within the BIM domain. Puolitaival and Forsythe carried out action research using Project-Based Learning (PBL) in an undergraduate context to support visual-spatial learning using BIM [11], who noted that while there was significant value to this approach, "challenges lay in the area of obtaining and developing appropriate BIM models as active resources well aligned with specified LOs ( . . . ). Simple models are clearly needed for demonstration purposes and for introductory exposure to 4D and 5D BIM tasks." Shen et al. [12] presented a similar project to that presented herein, whereby students used building energy analysis software with BIM using PBL and determined that this provided a good pedagogical approach for teaching sustainability within the building design and construction curriculum.

### 2.2. Constructivist Pedagogy

The theoretical premise of constructivism is that knowledge is "not transmitted directly from one knower to another, but is actively built up by the learner" [21]. Driscoll [22] summarizes the constructivist conditions for learning as follows: (1) Embed learning in complex, realistic and relevant environments; (2) Provide a social negotiation as an integral part of learning; (3) Support multiple perspectives and the use of multiple modes of representation; (4) Encourage ownership in learning; and (5) Nurture self-awareness of the knowledge construction process.

Constructivist theory has been considered in the BIM context by Elinwa and Agboola [13] who noted that virtual environments have a strong potential to create more richness in classroom interactions and found it to have significant potential for BIM education. Similarly, Eadie et al. noted that constructivism led to the concept of problem-based learning, which was noted to be ideal for teaching BIM [14]. Mathews [15] also engaged with constructivist pedagogy within the BIM context, applying Jonassen's [23] activity learning model components of (a) description of the physical, organizational, and socio-cultural elements of the problem; (2) related cases (related experience); (3) information resources; and (4) cognitive tools to help visualize, organize, automate, or supplant thinking skills to a fourth-year BIM course for an undergraduate Architectural Technology program, and found that this approach successfully equipped the students to serve in central problem-solving roles between disciplines.

The evaluation of constructivist learning focuses primarily on the effectiveness of the instructor and created environment, rather than measured student outcomes. Evaluation scales include the Constructivist Learning Environment Survey (CLES) [24], its adaptation to ICT teaching by Luan et al. [25], and Scale for Constructivist Learning Environment Management Skills (SCLEMS) by Yildirim [26]. The CLES measures learning environments across five scales: personal relevance of learning, the ability for students to express critical opinions, shared control in the learning, uncertainty, and student negotiations. The SCLEMS considers these aspects but maps them to seven factors with associated items for consideration, as summarized in Table 1. This latter scale was used as a framework for the evaluation of constructivist learning in this paper.

**Table 1.** SCLEMS Scale for evaluating Constructivist Learning Environments, summarized from reference [26].

| Factor | Items Considered |
|---|---|
| **Communication and Interaction** | Student opinions were solicited and considered<br>Students were encouraged to be enterprising<br>Students were encouraged to give independent decisions<br>Inter-student communication was encouraged<br>Students were included in rule-making and decision-making<br>Self-discipline and responsibility were encouraged in the students |
| **Relation Establishment** | Feedback given<br>Students asked to identify relations between their project learning and the real world<br>Open-ended questions asked<br>Students guided to draw conclusions and develop their knowledge |
| **Skills Development** | The following skills encouraged: Questioning, asking, and research; Critical and creative thinking; Problem-solving; information access and usage; determining the purpose of the assignment and how to execute it |
| **Time Usage** | Adequate time provided; students encouraged to use their time efficiently |
| **Assessment** | Different assessment techniques used and learning process considered rather than simply the results |
| **Learning and Teaching** | Concepts are the principal focus; various teaching methods used; learning activities were devised for active learning; learning centered around student interests and needs |
| **Learning Environment Organization** | Real-life problems considered; learning is made possible outside the classroom; various real materials and primary sources provided |

## 2.3. Experiential Learning

Kolb defined Experiential Learning as "*the process whereby knowledge is created through the transformation of experience*" [27]. This theory posits nine learning styles situated within a grid defined by two axes: concrete experience—abstract conceptualization, and active experimentation—reflective observation [28]. The former ranges from feeling vs thinking responses in the students, while the latter considers acting vs. reflecting responses. Within the domain of architectural education, experiential learning frequently consists of design research [29], design-build projects [30] and hands-on teaching using real-world equipment and technologies [31] to help students to physicalize and reinforce their classroom learning, thus permitting a much more comprehensive and applied grasp of the material. Virtual surrogates have also been used within this context and have shown similar promise [32].

The application of these learning styles has been considered and an experiential learning approach adopted by many educators teaching in the BIM context [16,17]. For example, Wang et al. [17] found that the use of virtual reality environments, including those integrating BIM, favoured more concrete experiential learning styles, namely the *Accommodator* (Feeling-Acting, at the far ends of the concrete experience and active experimentation scales) style.

Similar to experiential learning pedagogy, process-oriented instruction has been used in BIM pedagogy to foster and facilitate self-directed learning through participation, practice, peer-learning, group work, and inquiry.

To evaluate the overall effectiveness of a course or module in supporting experiential learning, Friedman et al. [32] recommend four sets of criteria corresponding to the four stages of experiential learning: (1) Experience—are the necessary tools provided to analyse current experiences?; (2) Reflection—are the tools adequate to help students reflect on their experiences?; (3) Conclusion—do the tools help students conclude upon experience?; and (4) Applying and planning—do the tools help to plan the next steps?

Gentry et al. [33] reviewed the evaluation of experiential learning and the measurement of tacit knowledge obtained and noted that Bloom's Taxonomy is widely used to measure experiential learning of individual students and provides a useful framework for this systematic review.

## 2.4. Evaluation Using Bloom's Taxonomy

Bloom's Taxonomy is a framework developed in 1956 [34] that consists of six categories of learning: Knowledge (recollection of facts), Comprehension (understanding of the material), Application (the use of abstractions), Analysis (identifying relationships between ideas and concepts), Synthesis (holistic knowledge creation), and Evaluation (making judgements regarding the value of materials and methods for given purposes). This approach has been broadly used for evaluating teaching across all disciplines, and has been applied to evaluating BIM learning outcomes. Sacks and Pikas [18] in their review of BIM process, technology, and application needs within the Construction, Engineering, and Management domain and development of a case study [19] determined that BIM education should be continuous, can enhance the learning of engineering concepts, and should incorporate real-world problems for analysis. Further, while BIM theory should be taught formally, they noted that self-learning is an effective means of learning the mechanics of BIM software. Arising from this review, Sacks and Pikas [18] recommended that Levels 1-3 (Knowledge, Comprehension, and Application) be the objectives of undergraduate education and that Level 4 (Analyze) constitutes best practice at the graduate level. Levels 5 and 6 were anticipated to only be achieved through work experience.

In 2001, Krathwohl [35] proposed a shift in Bloom's Taxonomy to consider not only the categories of knowledge but also the cognitive processes associated with each, responding to a shift from a static to a more dynamic teaching environment. This considered factual knowledge alongside conceptual, procedural, and metacognitive knowledge to develop the revised structure shown in Figure 1.

This latter structure was considered by Beach (summarized in reference [6]) in an Architectural Design context, who noted that the Drury University studio integration of BIM had been developed and subsequently refined to achieve Level 5 (Evaluate) and Level 6 (Create) of Bloom's taxonomy, and that significant revision of the course structure was required to achieve these levels.

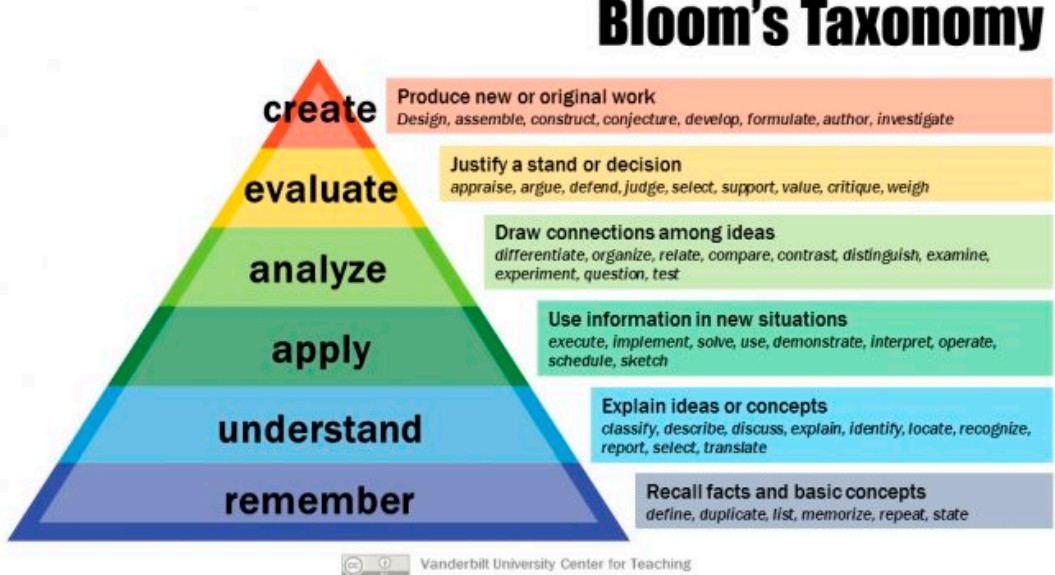

**Figure 1.** Bloom's Taxonomy (as updated by Krathwohl [35]), used under Creative Commons License from reference [36].

## 3. Case Study Methodology

A project was completed by 361 third-year Architectural Science students between 2015 and 2018. This Architectural Science program provides architectural education at an undergraduate level, complemented by technical courses within the building science (notably structures, HVAC, and building envelope analysis) and project management (project economics, cost estimating, scheduling, risk analysis, and construction management) domains. These are integrated in the third-year "Integration Studio" where building science and project management material are considered as part of a whole-building schematic design. This project linked the integrated design studio building conceptual design (massing) activity with the Project Economics course, incorporating both energy simulation and simplified cost take-offs (envelope only) to compare life-cycle energy and capital costs for various options. This project forms a component of the BIM Curriculum Integration Toolkit developed in 2015 as part of the Digital Education Curriculum described in detail by Hui et al. [37]. The objective of this toolkit was to expand BIM knowledge beyond the existing tutorials on *how* to use the BIM software, which was available to 2[nd] year students, and instead consider on *when and why* BIM should be adopted and the implications of BIM use on the project throughout its lifecycle. This focus on practice implications and holistic understanding aligns with the recommendations made by Sacks and Pikas [17] and was communicated through real project case studies, video interviews with architects and general contractors, project-specific video tutorials, and a comprehensive multi-media literature review.

### 3.1. Project Evolution

The data-driven design project was initially introduced in 2015, repeated in 2016, and developed through two further iterations in 2017 and 2018 to respond to both the lessons learned in previous iterations and adapt to a changing classroom environment. Figure 2 presents the three iterations evaluated in this case study. In each iteration, students created a number of different massing and envelope options and analyzed them either simultaneously (stacked vertically in figure, as in the first four options in Iterations 1 & 2) or sequentially (horizontal arrangement, as in the fifth option for Iteration 2 and options 2a, 2b, and 3 for Iteration 3) to inform a later design.

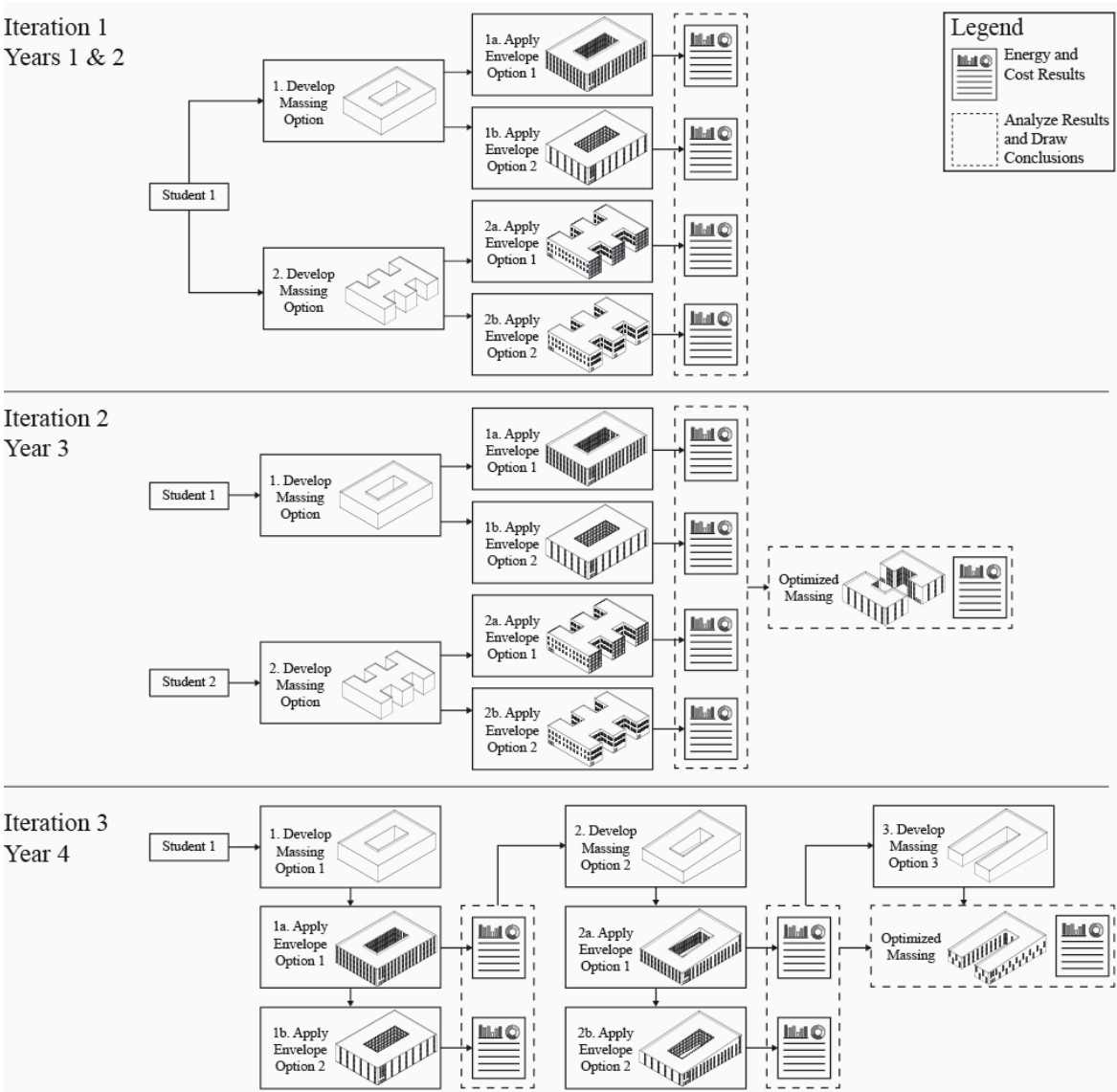

**Figure 2.** Overview of project processes in each iteration.

Of these iterations, the first (Years 1 and 2) was the most prescriptive, asking students to undertake a 2 × 2 factorial study considering two potential building shapes ("massings"), and testing each with two envelope variations. Prompts were given to students regarding these envelope approaches, giving them the choice to investigate: (1) two dramatically different envelope types, noting that opaque facades must include a reasonable amount of glazing for daylighting and views; (2) the same cladding material but dramatically different window-to-wall ratios (WWRs), with WWR consistent on each facade; or (3) consistent WWR but comparing even distribution (e.g., 40% glazing on each façade) to a distribution with the majority of glazing on the south facade and limited glazing on the east and west. Once created, students undertook two investigations for each model: a simplified capital cost estimate based on building envelope quantity take-offs and building lifecycle energy simulation using Green Building Studio (GBS) [38]. It was apparent early in Year 1 that the students' conceptual massing designs (from their concurrent studio course) were not developed enough to permit cost estimation beyond the envelope; and modeling to this level of detail would have been inappropriate at this stage. Instead, the building gross floor area was fixed at 5000m2 for all massing options to ensure consistency in design. To ensure appropriate simulation parameters were used, energy settings (system type, efficiency, and occupant loads) were prescribed in the project brief. Students synthesized their results

into a memo summarizing their justification for each envelope and massing considered, the results of the cost and energy analysis, an evaluation of which massing was most sensitive to changes in envelope, and their recommendations for the most cost-effective design. This pushed the students to achieve Level 4 (Analysis) and began to achieve Level 5 and 6 (Synthesize and Evaluate) of Bloom's taxonomy as defined for BIM education [18].

Based on the best results from these two years, and a student interpretation of the project highlighted in the following section, the 2017 project was revised to guide students to engage more fully with data-driven design concepts. In this new iteration, students performed this analysis in pairs, with each analyzing their studio project massing with two different envelope options, reflecting on their findings, and working with their partner to develop a new design drawing from their individual explorations to create a new massing concept with an optimized life-cycle cost. It should be noted that up until 2016, the GBS plug-in in Revit provided detailed heating and cooling load breakdowns, for example *window conduction* and *infiltration*, as standard outputs whereas from 2017 the software provided only end-use (e.g., fan, cooling, lighting) breakdowns.

In Year 4, extenuating circumstances, namely an uncharacteristically large cohort (20% larger than previous classes) and pressure to reduce the number of total assignments in each course led to a change in the course offering, whereby students selected three of five possible assessments. Due to limitations on group work total percentage in a course, and the presence of a large group project within the options, this project was modified slightly to be completed on an individual basis, rather than on a paired basis. This approach required that the project return to an individual assignment and a variation of the Year 3 approach was used whereby each student began with their studio conceptual massing and refined it four times, alternately changing the envelope and massing based on previous results. In this case, energy use intensity goals were also prescribed in the marking rubric for the first time to ensure that sustainability was a primary, rather than secondary, driver in the design. In all iterations of this project, Autodesk Revit [39] was used and a comprehensive set of tutorials was created to guide students through the mechanics of this process.

### 3.2. Evaluation Criteria

Two means of evaluating this assignment are discussed in this paper: the degree of student success in completing the assignment and their demonstrated understanding (Outcomes) and the extent to which the project achieved its pedagogical goals (Process). The former was formally evaluated on a student-by-student level using the rubric provided as a supplemental file, while the latter was reviewed retroactively as follows.

Model outcomes from Years 2–4 were evaluated in terms of the students' ability to (1) demonstrate appropriate modeling technique, (2) interpret energy results correctly, (3) evaluate costs correctly, and (4) integrate these findings to identify the best performing massing (Iteration 1) or decrease the life cycle cost and/or energy performance in a refined massing (Iterations 2 and 3).

The process is evaluated on an individual project basis and consolidated by year from Experiential Learning and achieved Bloom's Taxonomy levels, while a full-class perspective has been considered for the Constructivist evaluation. The criteria used for this evaluation are listed in Table 2. Note that it was impossible for a student to complete this project without achieving at least Level 3 of Bloom's taxonomy, thus Levels 1 and 2 are omitted from this table. Because a significant number of students engaged in some ways at Level 6 of the taxonomy without fully achieving it, an intermediate rating "Level 5+" was included in this table.

As noted in the introduction, Scale for Constructivist Learning Environment Management Skills (SCLEMS; summarized in Table 1) by Yildirim [26] was used to evaluate the effectiveness of the learning environment created from a constructivist perspective.

**Table 2.** Criteria for Pedagogical Evaluation.

| Pedagogical Lens | Rating | Criteria to Achieve Level* |
|---|---|---|
| **Experiential** | Low | Students did not show evidence of having engaged with the material; results were addressed but there was no discussion of how conclusions were formed nor was there evidence that the student showed adaptation in their approach from this assignment. |
| | Medium | Student showed a degree of adaptation of their approach, reviewing their results and attempting to explain their conclusion and build upon lessons learned from their previous analysis to guide later massing developments. |
| | High | Student showed a high level of engagement with the material, reflecting on their findings and clearly describing the insights and lessons learned from this experience. This can include testing ideas beyond those listed on the assignment and discussing their rationale for the development of their massing. Evidence of reflection on the findings to determine next steps is another indicator of this level. |
| **Bloom's Taxonomy** | Level 3—Apply | Student executed the assignment, applying the project instructions to their massing. ("Energy study provides output for both glazing iterations with minimal text describing it") |
| | Level 4 - Analyze | Student was able to compare results from the various massings and draw conclusions from their investigations. ("Analysis of the data is provided and clearly explains the findings of the results and any issues arising in the process.") |
| | Level 5—Evaluate | Student was able to judge between the results, select the massing option (of those created) with the best performance, and articulate its superior performance with some degree of insight as to what caused this success. ("Student demonstrates awareness of implication of their results for massing selection.") |
| | Level 5+ | The student was able to clearly articulate why certain massings performed better than others and speculated how they could adapt their design to create an improved design based on their understanding, but failed to fully execute the resultant design. |
| | Level 6 - Create | The student was able to draw from their analysis of earlier massings to improve the performance of a final massing. ("Student demonstrates awareness of implication of their results for design development and/or massing selection.") |

* Where evaluated, rubric text is n quotations. This is evident in Bloom's taxonomy [18].

## 4. Case Study Results

Of the 361 projects completed over the four year period, the 261 from Years 2-4 were evaluated in this research as the Year 1 work was submitted in hard copy and returned to students. The following evaluation thus considers 100 projects (400 models) from Year 2, 95 projects (235 models) from Year 3, and 60 projects (266 models) from Year 4. Table 3 summarizes the pedagogical findings by year for these projects. Selected models are shown in this section to highlight key outcomes for each year. As these images cannot be provided at a scale to ensure text legibility, a supplemental file has been provided with the transcription of each presented project's written analysis.

**Table 3.** Summary of Results by Year.

| Criteria | Category | Year 2 | Year 3 | Year 4 |
|---|---|---|---|---|
| **Experiential Learning** | Low | 17% | 9% | 32% |
| | Medium | 71% | 80% | 50% |
| | High | 12% | 11% | 18% |
| **Bloom's Taxonomy Maximum Level Achieved** | Level 3 | 4% | 0% | 8% |
| | Level 4 | 42% | 17% | 30% |
| | Level 5 | 31% | 54% | 35% |
| | Level 5+ | 21% | 17% | 17% |
| | Level 6 | 3% | 13% | 8% |

It is immediately apparent from this table that Year 3 was the most successful overall. From an experiential learning standpoint, Year 4 showed the worst performance, due in part to variability between classes—this cohort was noticeably less engaged in all classes, but was exacerbated by the shift from team to individual work, thus not requiring the students to engage with one another in this project. Despite this decreased experiential learning, the Bloom's taxonomy score increased from the previous individual work iteration (Year 2). Based on these results, the benefit of the paired work in conjunction with a more data-driven design approach is recommended. Detailed discussions of the student outcomes and pedagogical evaluations by year are presented in the following sections, supplemented with excerpts from sample projects.

### 4.1. First Iteration (Year 2)

On average, students were able to achieve a 26% reduction (interquartile range (IQR):15%–34%) in the building life cycle cost. Of the models created, 94% were correct and had expected energy performance results while 25 (5.9%) had the following issues: excessively high underground losses likely due to incorrect ground plane definition (16 models), zero window solar or conduction gains due to incorrect window family definitions (5 models), highly variable occupant densities due to inconsistent energy settings (2 models), and extremely high roof losses due to inconsistent material assignments (2 models). The student work from Year 2 demonstrated an understanding of the academic material, in which students drew insight from their work. From student feedback there appeared to be hesitation in the initial approach of the project, as the majority of the criteria in the assignment was new material that the students did not experience in previous classes and projects. However, this did not pose any problem for the students to complete the assignment successfully. What was particularly noticeable in the year was the student's ability to investigate these new topics and pose critical thinking of the energy analysis and cost estimates that derived from the building massings. Students were proactive with providing multiple potential causes for their results, which demonstrates a broader understanding of the course curriculum. Even at times when the data was incorrect, some students were able to identify these anomalies and reflect on them. The following student statement demonstrates this awareness: *"( . . . ) there is an odd condition where there is more of a demand for energy in both fuel and electricity for the exterior window ratio of 0.20 over 0.80 ( . . . ) The only possibility ( . . . ) is that walls are less insulated than the glazing, or if the HVAC unit performs poorly and requires the use of glazing to mediate the interior temperature to a level that would not need additional energy ( . . . ) However, due to the unlikeliness of these situations, it might be more logical to establish that there is an error within the Revit model itself, or Green Building Studios."* By completing the assignment, the students exposed themselves to new design and evaluation approaches for their building massings and embraced these concepts for potential use in future academic projects.

The project in Figure 3 demonstrates student engagement and critical thinking in the analysis portion of the assignment. The student explored variations in the WWRs, in addition to the amount of compactness in the overall building form. From the energy analysis, the student observed and noted the impact of envelope compactness on energy performance as well as the significant costs associated with high glazing percentage in cold climates. This recognition of daylight-heating/cooling load trade-off of high amounts of glazing is representative of the student experience as a whole; 96% of students came to similar conclusions and were able to improve building performance by reducing WWR (96%) and/or specific glazing orientation (44%). In addition, 53% of students used this exercise to quantify the energy savings associated with increasing the compactness of the built form, and were able to use this insight to further refine their schematic design.

This assignment created a reasonably good constructivist learning environment. There was a moderate degree of Communication and Interaction; while the brief was highly prescriptive, the choice of massing options presented students with flexibility to tailor their project to their interests. Students received feedback, were explicitly encouraged to draw real-world conclusions from their analysis and develop their knowledge, and a significant percentage of marks were assigned to the

open-ended questions, which all address Relation Establishment. The students developed significant research, critical thinking, and information access skills, and adequate time was provided for this project. Because of the structure of the rubric, both the actual results and the students' interpretation thereof and extrapolation to real-world scenarios were marked, allowing the instructor to assess both the learning process and the outcome. From a learning and teaching standpoint, the concept of cost analysis and energy simulation *as design tools* was stressed and the learning activity was designed to promote active learning by students. Finally, the learning environment was organized such that learning was necessary outside the classroom, supported by video tutorials accessible on-demand by students, and was directly tied to a real-life problem. Despite the degree of prescription in the project brief, one student opted to modify his assignment to best use BIM as a data-driven design tool and achieve significantly improved results. The resultant project is highlighted in Figure 4, and served as the inspiration for future iterations of the project to guide other students towards similar analysis.

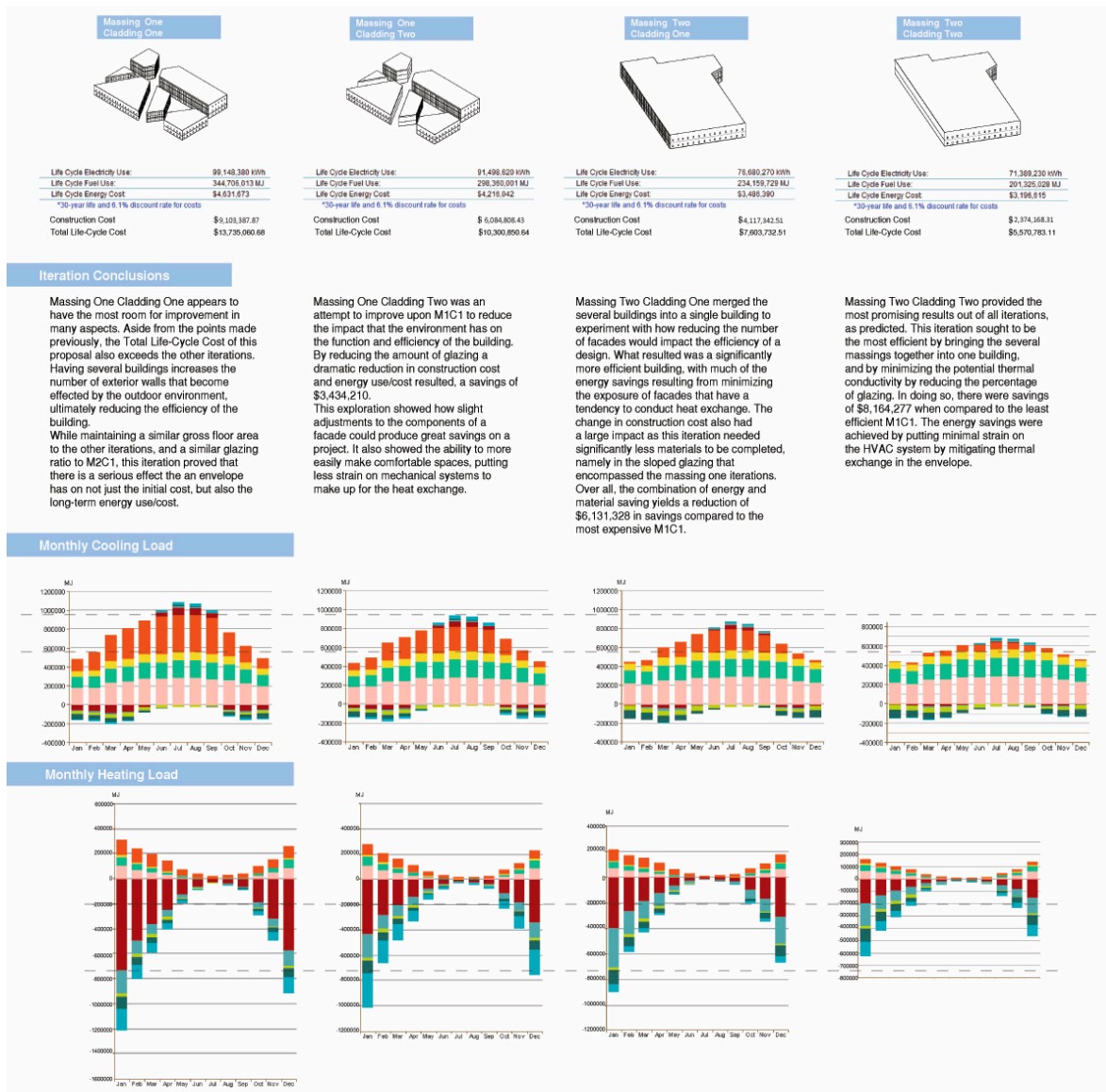

**Figure 3.** Example of a Factorial Investigation from Year 2 (image used with permission). Text transcription is provided in the Supplementary material.

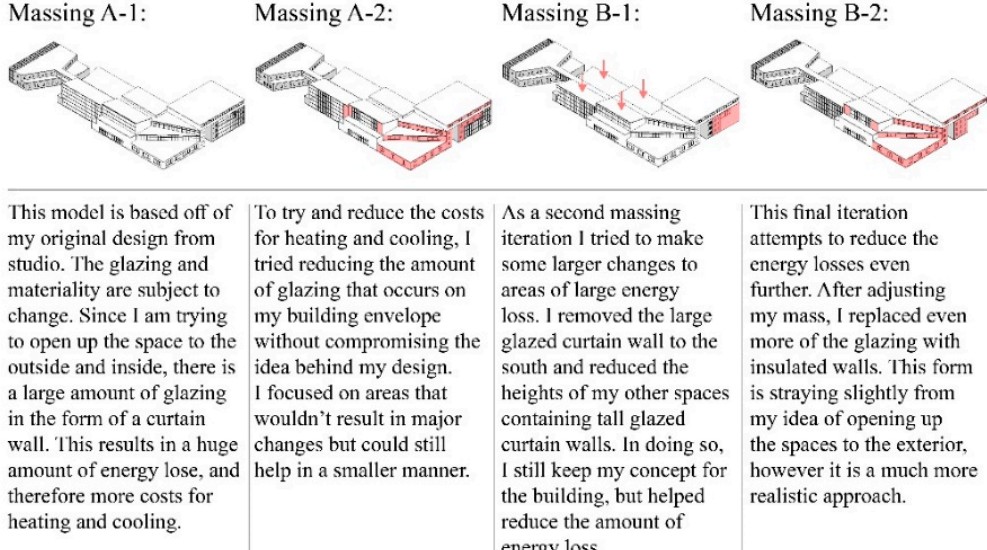

Massing A-1:

This model is based off of my original design from studio. The glazing and materiality are subject to change. Since I am trying to open up the space to the outside and inside, there is a large amount of glazing in the form of a curtain wall. This results in a huge amount of energy lose, and therefore more costs for heating and cooling.

Massing A-2:

To try and reduce the costs for heating and cooling, I tried reducing the amount of glazing that occurs on my building envelope without compromising the idea behind my design. I focused on areas that wouldn't result in major changes but could still help in a smaller manner.

Massing B-1:

As a second massing iteration I tried to make some larger changes to areas of large energy loss. I removed the large glazed curtain wall to the south and reduced the heights of my other spaces containing tall glazed curtain walls. In doing so, I still keep my concept for the building, but helped reduce the amount of energy loss.

Massing B-2:

This final iteration attempts to reduce the energy losses even further. After adjusting my mass, I replaced even more of the glazing with insulated walls. This form is straying slightly from my idea of opening up the spaces to the exterior, however it is a much more realistic approach.

**Figure 4.** Factorial Investigation—Case Study #3 (Used with permission).

*4.2. Second Iteration (Year 3)*

On average, individual students were able to achieve a 14% reduction (interquartile range (IQR): 5%–18%) in the building life cycle cost. Of the models created, 89% provided reasonable results, based on the evaluation of calculated Energy Use Intensity (EUI), end-use breakdowns, and fuel use ratios. Due to the focus on cost for the final optimized design, only 27% of students were able to achieve an improvement in their EUI, although 76% were able to decrease their lifecycle cost. The modeling was generally well done, with 91% of students having no notable modeling issues. Of the remainder, 11 models showed excessively high underground losses likely due to incorrect ground plane definition, 9 showed excessive energy consumption due to infiltration, and two showed irregular energy consumption due to poor modelling of the building envelope (2 models).

The student work showed a significant increase in engagement with the material than in the previous year, and the number of students demonstrating moderate or high levels of experiential learning increased from 83% to 91%. The proposed design iterations that the students made were more deliberate with some being structured as practical design problems that needed solutions, rather than in the previous project iteration, in which students simply compared cost and energy data of different design iterations. This could be the result of the pairing, as students could share and reflect on their learning together to come to more informed decisions. Additionally, having provided the structure of the project in its second iteration, the creation of a final optimized massing allowed students to apply themselves at a higher level in which they could make educated decisions in improving the performance of their new massing. The following student quote demonstrates Bloom's Level 5+ evaluation: *"( . . . ) by decreasing the number of windows, there is an increase on the electrical load due to less natural lighting. In massing 1, the annual electrical for lighting was 23%, but raised to 30.1%. ( . . . ) Although the second massing performs better in the overall annual and lifecycle energy, due to the higher electrical costs it makes it have a higher cost."* By defining the cause of the fluctuation in data from the electrical load, the student is demonstrating their ability to analyze (Level 4) and evaluate (Level 5). To receive a rating of Level 6, a student would have to exceed the previous example and further emphasize reflection from their analysis and evaluation. An example of this is as follows: *"This transparent pathway is a main feature in my building. ( . . . ) This pathway will end up costing me much more in heating during the winter and cooling during the summer. From this analysis I realize I must look into either extensively shading the center path or removing it and focusing on northern glazing. Not only is the glass enclosed pathway a main source of heat loss, it also cost much more to construct than a typical wall assembly."*

The sample project progression illustrated in Figure 5 shows the series of project investigations undertaken by a pair of students and clearly shows how they drew conclusions from each of the preliminary massing investigations (Designs A and B and their modifications) to inform an optimized design achieving 30% life cycle energy savings and 9% initial capital cost reduction, resulting in a 20% life cycle cost savings. From the lessons learned of the previous massings, the students focused on reducing the building surface area, rearranging the glazing to decrease the amount of openings and focusing on elongating the building along the East-West axis to increase solar gains from the south for the final optimized massing.

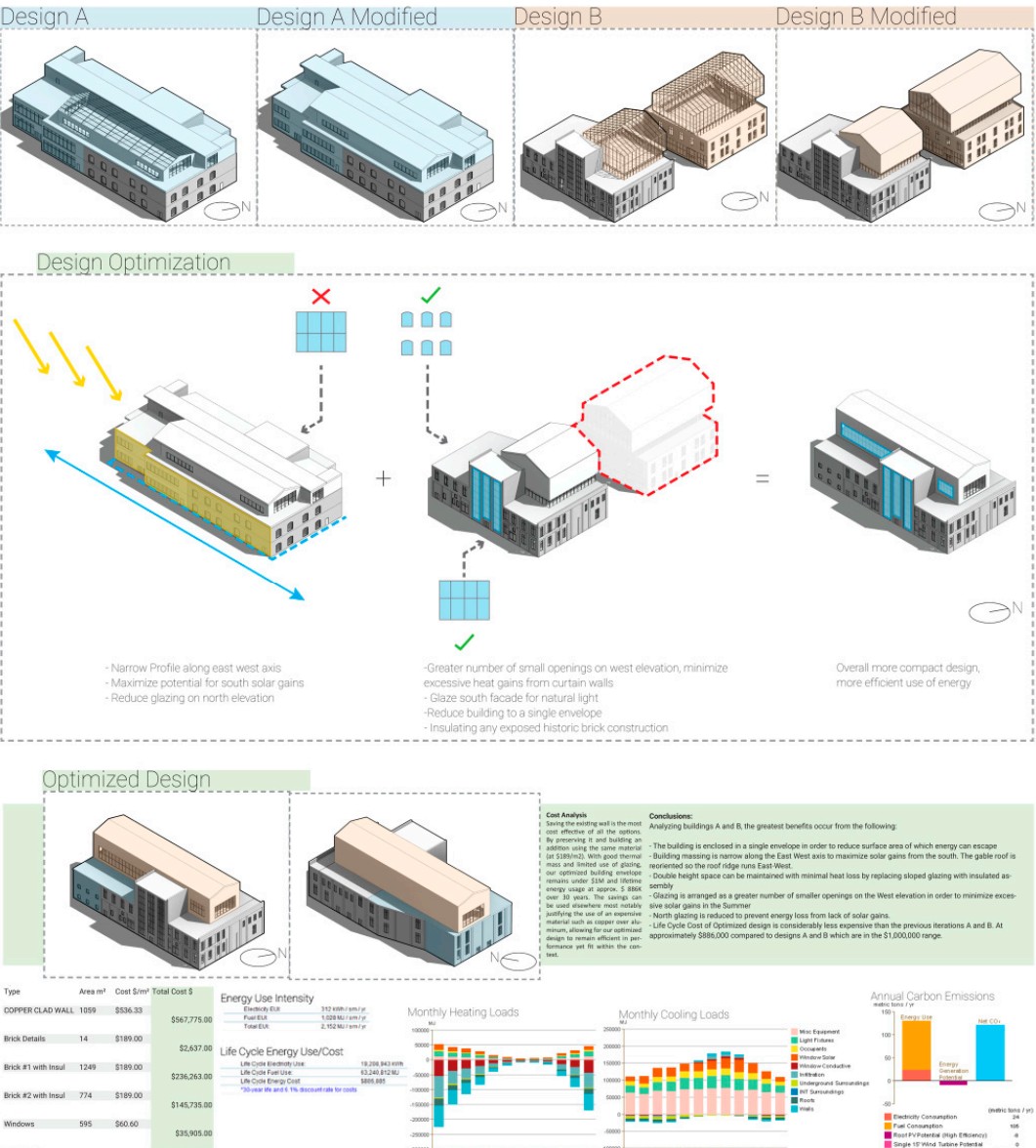

**Figure 5.** Year 3 Sample Showing Design Development Resulting in an Optimized (Life Cycle Cost + Energy) Design. Text transcription is provided in the Supplementary material.

The second sample project illustrated in Figure 6 shows a pair of students explore the different repercussions of their glazing orientation. The first massing iterations explore the energy load and construction cost of incorporating a skylight along the length of the building, while the second massing iterations explores the reduction of southern glazing to decrease heat loads. From the lessons learned in both design explorations, the students were able to formulate a new optimized massing that focused

glazing orientation on particular facades to maximize views to the exterior while achieving a 5% reduction in EUI in comparison to the previous iterations.

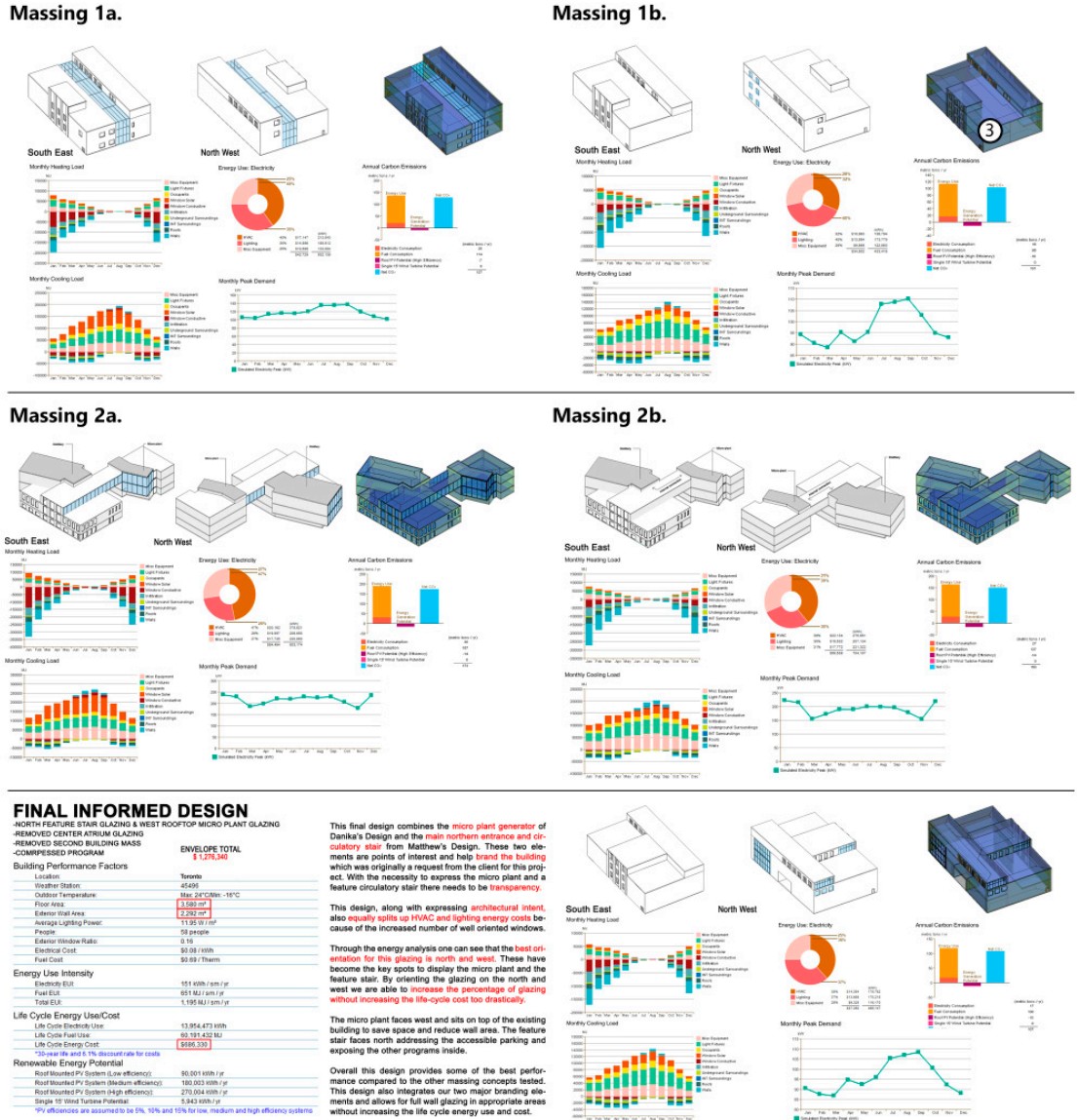

**Figure 6.** Year 3 Sample Showing Design Development Resulting in an Optimized (Life Cycle Cost + Energy) Design.

An analysis of the constructivist learning environment engendered by this assignment found similar results to Year 2 in all categories except that peer-to-peer interaction was innately required in this project, increasing the degree of Communication and Interaction significantly from the previous year. The decrease in prescriptiveness in the brief—students were encouraged to explore any changes to their studio massing they felt appropriate as their second option for comparison—further provided students with flexibility to adapt this assignment to align with their own interests and curiosity. While the discussion regarding the appropriateness of GBS was moved to an in-class discussion and not present in the written report, the rubric rewarded evidence of critical thinking and reflection in the massing development, again allowing the assessment of both the learning process and the final outcome.

### 4.3. Third Iteration (Year 4)

The extenuating circumstances in Year 4 provided an (unplanned) opportunity to investigate the impact of teamwork on the project outcomes. This was the first year where there were explicit EUI requirements as well as a stated goal to reduce life-cycle cost, as there had been multiple students in previous years who had focused exclusively on total cost savings, at times to the detriment of energy savings. In their first four massing iterations, 83% of students achieved EUI savings with an average of 28% (IQR: −5%–23%) reduction in the first optimized iteration, 27% reduction (IQR: −4%–24%) in the second optimized iteration, and a 33% reduction (IQR: 36%–69%) in the third. Just over half (52%) of the students noted the daylight-heating/cooling load trade-off. The majority of students were able to improve building performance by adjusting the window-wall ratio (80%) and/or increasing envelope R-Values (48%). In addition, 46% of students were able to identify the correlation between energy savings and increasing the compactness of their final form. Of the models developed, 97% provided reasonable results based on evaluation of EUI, end-use breakdowns and fuel use ratios. Despite these good modeling results, many students did not complete the project, stating only their concept of what might make for an optimal design, rather than creating a fifth iteration, resulting in Level 5+ rather than Level 6 outcomes for these individuals. This may have been the result of the change to individual work, which limited peer-peer teaching and eliminated the evaluation of peer-generated massing iterations. In contrast to the previous year, there was a noticeable reduction in the creativity of the design problems and solutions, rendering some of the projects from the third iteration similar to the expectations of the first project iteration. While variability between class performance is expected and a broader evaluation of this cohort's performance compared with previous years explains some of the decrease in results, it is clear that the change to individual-only work further enhanced this issue. It is therefore strongly recommended that this project integrate the paired design refinement in the future.

Despite these challenges, there were good examples of students engaging with the project advanced levels. For example, one student demonstrated a high level of analysis, synthesis, and evaluation of the factors impacting energy performance and conjecture of how to improve it as-follows: "( . . . ) *the energy intensity of the building to be 198kWh/m$^2$\*yr) of that 198, ( . . . ) 40% is spent on space heating. ( . . . ) The glaring issue of the massing is its large surface area due to the roofs and many voids throughout, creating a lot of surface area for the building, if some of these voids were to be enclosed, the spaces can maintain the same program and be used all year round, reducing the surface area reduces the area in which heat exchange between inside and outside can occur*." By reviewing the EUI and posing the consequences associated with it, the student is demonstrating the ability to analyze (Level 4) and evaluate (Level 5), moving into Level 5+ with their conjecture and identification of new approaches to address the surface area.

A sample student project achieving level 6 is shown in Figure 7. Based on preliminary massing investigations, the student determined that "*The key observation for this project was the impact of surface area to volume ratio, and the impacts of these factors in steady state heat conduction. The next steps for this project would to also design the building in such a way that would mitigate effects such as wind washing or air leakages, it's possible for a textured surface area, at a scale smaller than those provided in these massings, to serve as wind breakers to ideally reduce exterior wind-induced pressure profiles*". By applying this insight, this student achieved a 32.7% reduction in EUI (to 142 ekWh/m$^2$/yr) compared to the previous massing iterations that were achieved.

In the second example project illustrated in Figure 8, another student explored the consequences for modifying the window-wall ratios along the West and South façade, along with a modification on the atrium skylight. From the initial design iterations, the student was able to conclude that reducing unnecessary glazing around the building, and focusing it on the primary portions of the façade significantly reduced their EUI. By applying these lessons learned and even reducing glazing on the northern and easterly facades, the student was able to achieve a 76.2% reduction in overall EUI compared to the initial massing option while still retaining their design intent.

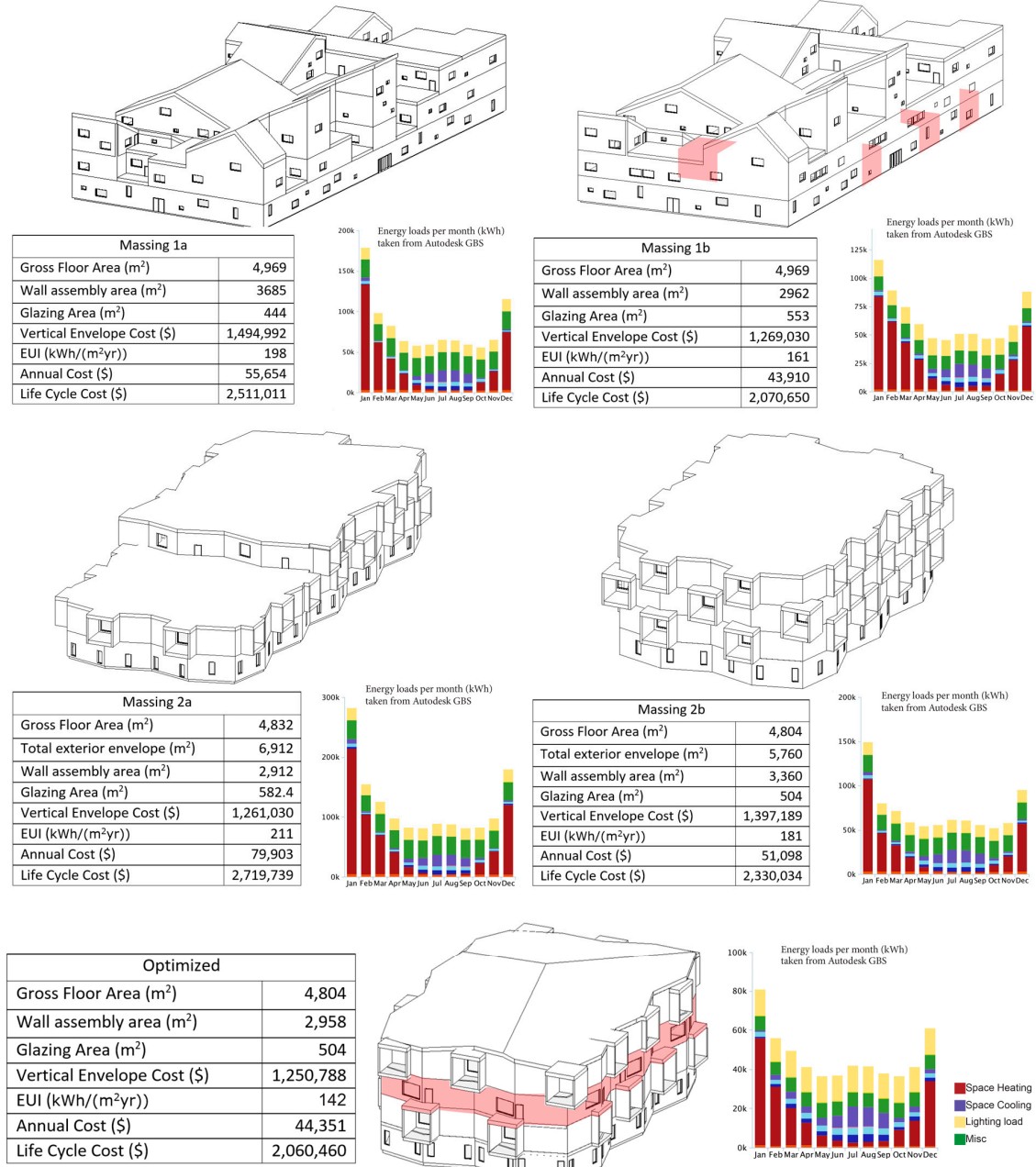

**Figure 7.** Year 4 Example showing refinement and significant energy reduction (used with permission; name withheld)).

Despite the efforts to further loosen the project brief and enable a high degree of student-led learning, the lack of peer-to-peer interaction significantly decreased the success of this project from a constructivist perspective environment. While the strongest students performed exceptionally well and were able to take full advantage of the flexibility afforded within the project, weaker students suffered significantly from the lack of peer-to-peer interaction prescribed in the previous iteration. In order to achieve the best learning outcomes for all students, it is therefore critical that this project incorporate the peer-teaching and joint design element.

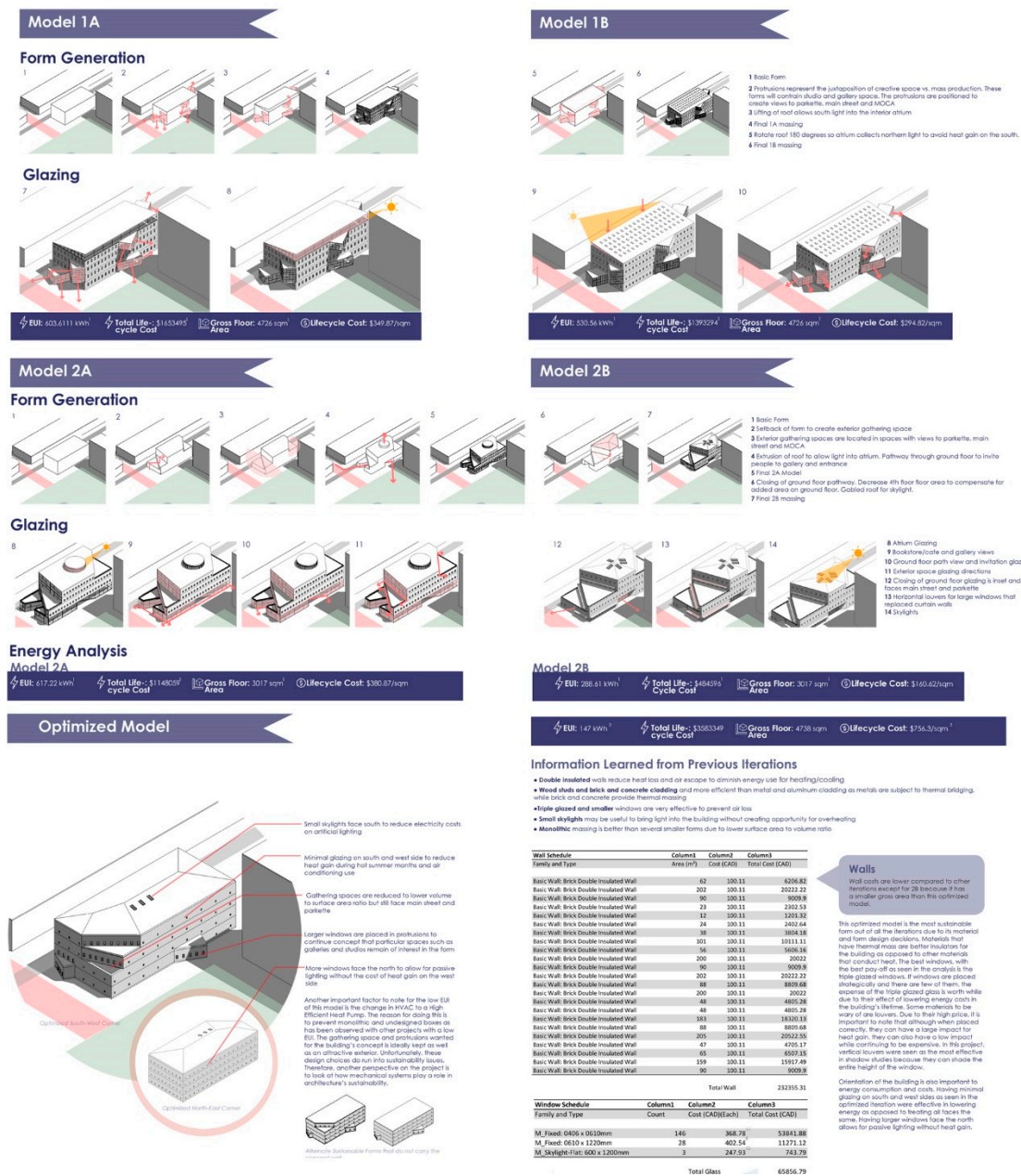

**Figure 8.** Year 4, Example 2 showing progression (Used with permission (name withheld)).

## 5. Opportunities for Advanced Project Implementation

One student's work after this course indicates the potential for this approach to support more advanced courses or Capstone projects. In 2017, one student entered a national competition with the criteria to design an environmentally sustainable building. From the lessons learned in the first iteration of the study, the student undertook the initiative to use a computational approach to evaluate self-designed building massings to inspect the ramifications on the building's heating and cooling loads and its overall energy usage. Two types of open-source software, Ladybug and Honeybee [40] for Grasshopper (a plugin for Rhinoceros [41]), connected with simulation engines Energy Plus [42], Radiance [43], Dayism [44], and OpenStudio [45], were used in this design exploration. By applying the data-driven approaches used in the project, the student evaluated the surface area of the roof(s) of each building to determine photovoltaic generation potential and compared it with the building energy consumption to achieve a net zero building for a pre-selected site. This analysis is summarized in Figure 9.

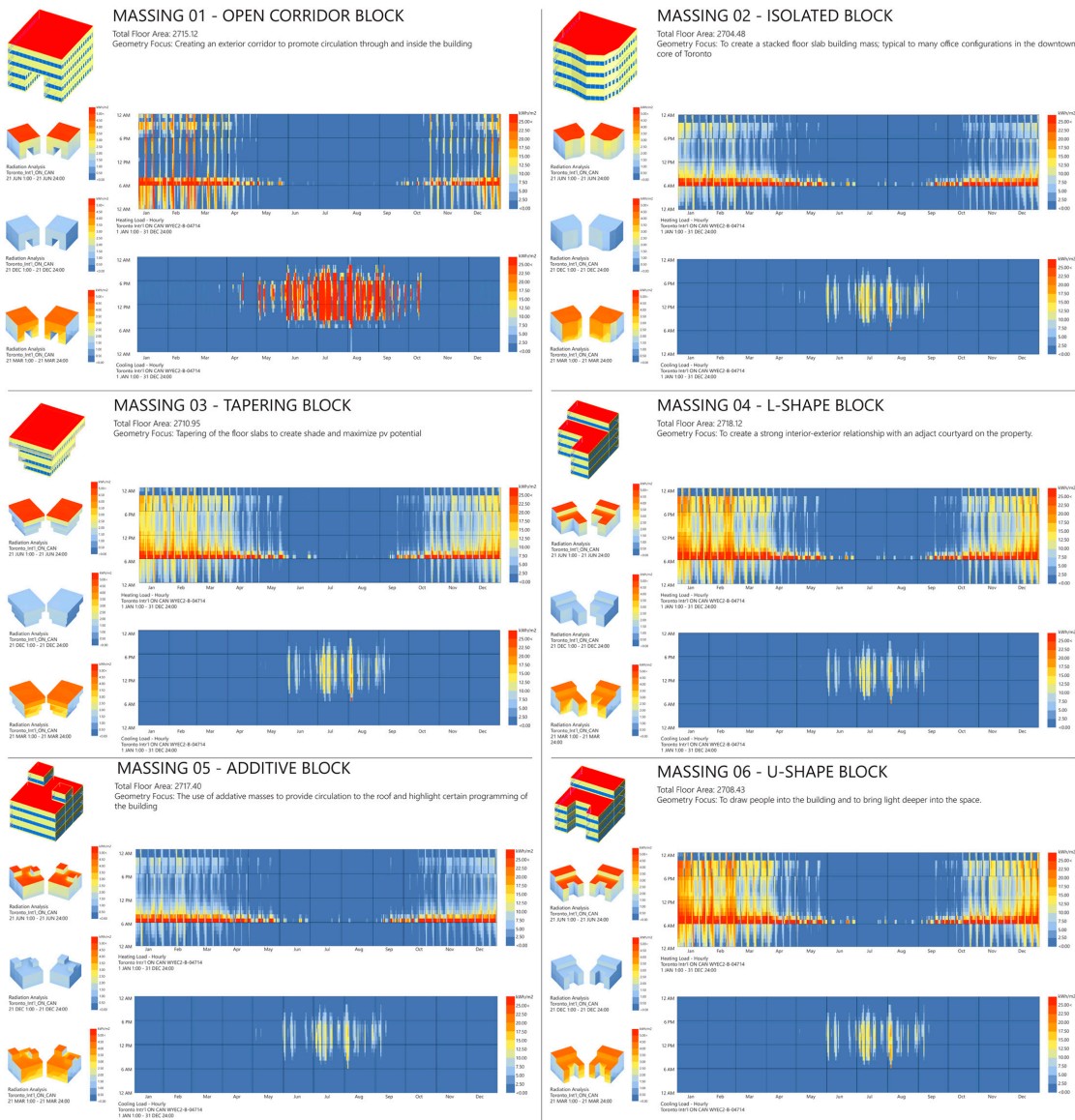

**Figure 9.** Competition entry: Radiation analysis evaluating heat gains on different building geometries.

This application of the learning had the student engage more comprehensively in a parametric design, creating and evaluating hundreds of micro-iterations by varying building characteristics using slider elements and observing the impact on building performance. For example, the window to wall ratio on each façade was varied, permitting observation of how this parameter impacted heating and cooling loads in real time. This encouraged broader experimentation with the building forms and provided a more dynamic learning experience, but required substantially more faculty mentoring than the presented project. The grasshopper script used to create this model was genericized and provided as an alternate means of completing the undergraduate project in Year 3, however the students were intimidated by this degree of analysis and preferred to use the project as initially developed.

## 6. Conclusions

The past four years of this project have provided significant insight into data-driven design projects as a means to elevate BIM capacity at the undergraduate level. The results demonstrate how an experiential approach to BIM through iterative design-analysis-synthesis cycles permit such student engagement in a sophisticated manner. Overall, the project resulted in significant capacity being

gained by the students to model, simulate, and evaluate design iterations by leveraging data generated using BIM, providing both an improved understanding of how design decisions affect capital cost and energy use, and promoting critical thinking through the design process. Inherent to the project was the development of high-quality BIM models and advanced BIM skills such as cost estimation and in-situ energy simulation.

Considering the student performance outcomes, the overall model quality was quite high, with fewer than 10% demonstrating modeling issues that resulted in unrealistic energy performance, and the review of the underlying reasons for these issues—poor wall family definitions and geometric errors—has informed the creation of additional learning resources. From an environmentally sustainable design perspective, the student learning outcomes were good, with the vast majority of students gaining insight into key cold-climate design concepts. These included the daylight-heating/cooling load trade-off of high amounts of glazing, which led over 90% to achieve improved energy performance through changes to glazing percentage and/or orientation, and an appreciation for the reduced sensitivity to climate afforded by more compact massings (approximately 50% of students). Student feedback obtained through surveys and course evaluations demonstrates that this project is effective to both equip students with advanced BIM skills, but also encourage them to synthesize a broad range of data generated through simulation to refine and develop their designs to improve sustainability. Cost take-offs were consistently well executed and integrated into a coherent capital cost estimate, and this aspect of design was well-considered in the project evaluation and (Years 3 and 4) refined designs.

As demonstrated in the selected examples and statistical analysis presented, students consistently demonstrated an increased understanding of building physics as well as project economics through their engagement with this project, effectively using BIM to undertake simple analysis and evaluate this analysis to make informed decisions to refine their designs. Students consistently demonstrated the use of BIM for analysis, synthesis, and evaluation, thus beginning to achieve higher educational learning outcomes recommended by Sacks and Pikas [18]. Based on the success of this project, the use of GBS for design analysis and refinement have been regularly integrated into the design student course in the winter term, providing students with additional opportunity to engage with simulation to evaluate design alternatives and achieve improved building performance. From a sustainability perspective, this was a valuable exercise, and provided students with insight into how building massing and envelope decisions impact energy performance. While earlier iterations of the project focused on life cycle cost savings, energy consumption was reduced by an average of 16% (IQR = 10%–27%) in Years 2 and 3 and increased to an average of 50% (IQR of 36%–39%) in Year 4 when energy performance was stated as a key requirement of the project.

Pedagogically, the second iteration of the project provided the best results overall, and the benefit of the paired work to develop the final refined design is evident from an experiential learning and constructivist perspective. The students consistently demonstrated keen levels of engagement through the execution of the project, reinforced by the creativity in framing their design problems. Further, the highest levels of achievement within Bloom's Taxonomy were achieved during this iteration. However, because this project draws from both experiential and constructivist learning theories, this project continues to evolve. The focus on EUI as well as lifecycle cost in the third iteration reinforced the sustainable design element of the project more substantially than the focus on lifecycle cost only, and therefore a minor revision to the project workflow from Iteration 2 incorporating this element forms the basis for future work. The project workflow for this recommended implementation is demonstrated in Figure 10.

Because a limitation of this study is that the single-program context limits generalization, the comprehensive project brief has been provided as a supplemental file to permit interested educators to use this project in their own teaching. Given that this approach aligns well with recognized best practice and pedagogical recommendations, it is hoped that others will choose to replicate this project and that findings be shared in order to evaluate the applicability of this approach across geographic and pedagogical contexts (architecture, architectural engineering, building engineering, building science,

or other related fields) and thus confirm the validity of this approach or inform modifications to further enhance the learning outcomes.

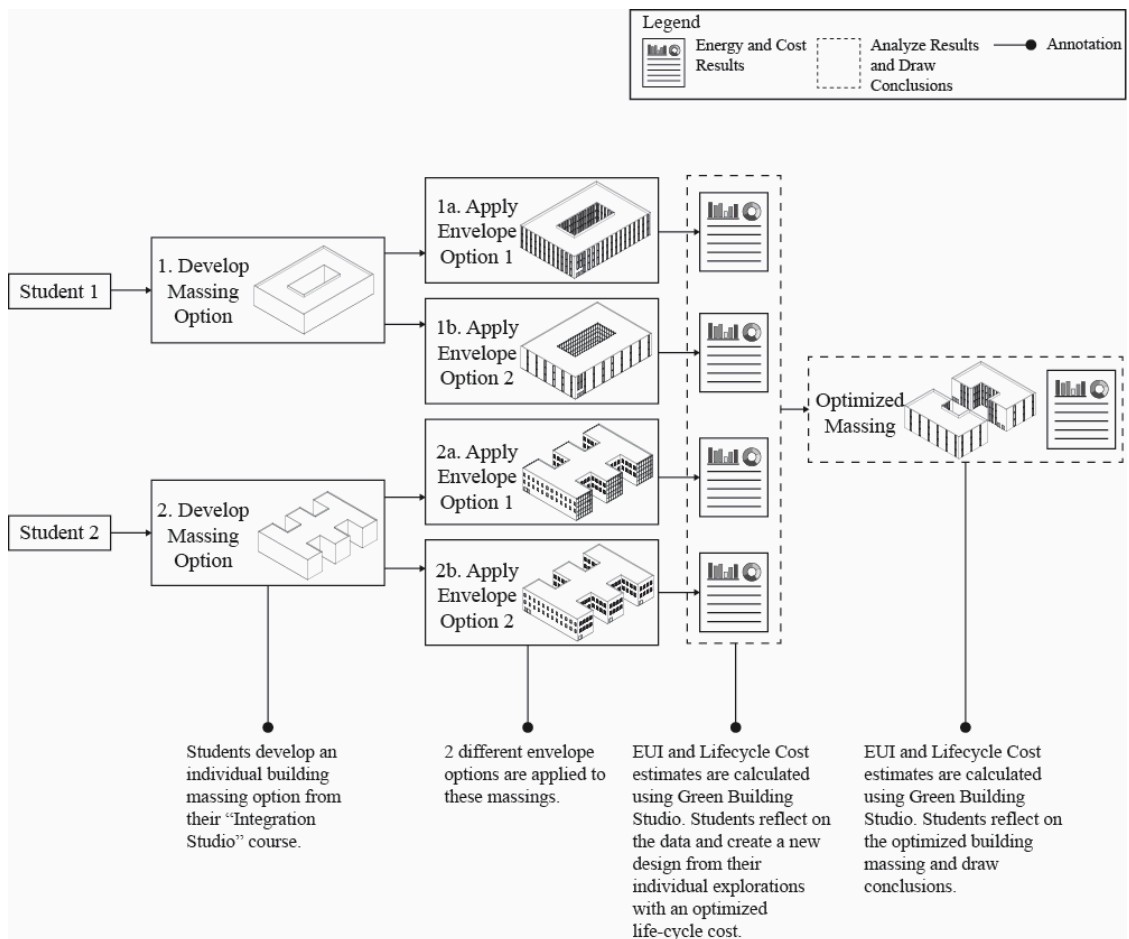

**Figure 10.** Project workflow for recommended project deployment.

**Supplementary Materials:** The following are available online at http://www.mdpi.com/2075-5309/9/5/103/s1.

**Author Contributions:** Conceptualization, J.J.M.; Data curation, J.B.; Formal analysis, J.B. and J.J.M.; Investigation, J.B.; Methodology, J.J.M.; Project administration, J.J.M.; Resources, J.J.M.; Software, J.B.; Supervision, J.J.M.; Validation, J.B. and J.J.M.; Writing—original draft, J.B.; Writing—review & editing, J.J.M.

**Funding:** This research was funded by Ryerson University's Learning and Teaching Enhancement Fund and the "Fall/Winter Work Study Research Assistant Program".

**Acknowledgments:** The authors thank the students who consented to the inclusion of their work within this paper.

**Conflicts of Interest:** The authors declare no conflict of interest.

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
