# Peer review of "Data-Driven Design as a Vehicle for BIM and Sustainability Education"

_buildings, doi:10.3390/buildings9050103_

Round 1

Reviewer 1 Report

0. My comments are made from my experience on pedagogical strategies and Architecture.

1. Am I wrong or it is not easy to find the 34 references in the main text? For example, references in first paragraph of Chapter 2 are needed. Same similar happens across the text.

2. More references about antecedents on how to improve Architecture learning would be welcome. Some suggestions not necessarily to be included but which could open new lines to authors:

“Emergency lighting cabinet for fire safety learning”. Case Studies in Fire Safety, Volume 3, May 2015, Pages 17–24. ISSN: 2214-398X. DOI:10.1016/j.csfs.2014.11.001.

"Building services cabinets as teaching material in a degree in architecture". European Journal of Engineering Education, Vol. 38, No. 5, 2013, 468–482. ISSN 0304-3797 (Print), 1469-5898 (Online). DOI:10.1080/03043797.2013.833176.

3. Table 2. Factors order justification? Some of them starting with capital and other no.

4. First paragraph of 3.1. Project Evolution: Good work.

5. Page 7. Line 212. "Extenuating circumstances". Please, explain.

6. Table 3. Bloom, not bloom.

7. Is it possible to create a new figure explaining the work flow during the project? There is a lot of information, but it requires to read several time the information to understand properly. That it is, if I am a professor who wants to repeat this experience, what it is the procedure to follow? 

8. Please, in conclusions explain future works, or is this a close way in teaching? I think not, so an explanation is needed.

Author Response

We have responded to the reviewer comments in blue text below:

Reviewer #1

Comments and Suggestions for Authors

0. My comments are made from my experience on pedagogical strategies and Architecture.

We thank the reviewer for this context and have found their helpful to be extremely helpful.

1. Am I wrong or it is not easy to find the 34 references in the main text? For example, references in first paragraph of Chapter 2 are needed. Same similar happens across the text.

 We have reviewed the references and corrected both the reference and updated the numbering for improved clarity.

2. More references about antecedents on how to improve Architecture learning would be welcome. Some suggestions not necessarily to be included but which could open new lines to authors:

“Emergency lighting cabinet for fire safety learning”. Case Studies in Fire Safety, Volume 3, May 2015, Pages 17–24. ISSN: 2214-398X. DOI:10.1016/j.csfs.2014.11.001.

"Building services cabinets as teaching material in a degree in architecture". European Journal of Engineering Education, Vol. 38, No. 5, 2013, 468–482. ISSN 0304-3797 (Print), 1469-5898 (Online). DOI:10.1080/03043797.2013.833176.

Thank you for these suggestions. We have expanded the literature review to consider the diversity of experiential learning within architecture using design-build projects, hands-on cabinets such as these, and virtual surrogates as a means to better reinforce curricular content. This additional discussion has been included in lines 119-123.

3. Table 2. Factors order justification? Some of them starting with capital and other no.

We listed experiential learning prior to Bloom’s taxonomy to mirror the structure of the introductory material, and showed increases from low to high and Level 3 to Level 6 to show the progressive requirements for students to achieve the higher level. All have now been capitalized for consistency.

4. First paragraph of 3.1. Project Evolution: Good work.

Thank you very much for this encouraging remark.

5. Page 7. Line 212. "Extenuating circumstances". Please, explain.

We have added the following text to the manuscript (lines 228-237): “extenuating circumstances, namely an uncharacteristically large cohort (20% larger than previous classes) and pressure to reduce the number of total assignments in each course led to a change in the course offering, whereby students selected three of five possible assessments. Due to limitations on group work total percentage in a course, and the presence of a large group project within the options, this project was modified slightly to be completed on an individual, rather than paired, basis.”

6. Table 3. Bloom, not bloom.

This has been corrected as noted.

7. Is it possible to create a new figure explaining the work flow during the project? There is a lot of information, but it requires to read several time the information to understand properly. That it is, if I am a professor who wants to repeat this experience, what it is the procedure to follow? 

This had been the intention of Figure 2, however this comment has helped us to understand that this was not successful as-presented. In response to this comment, we have modified the text adjacent to Figure 2 to better explain the figure (lines 191-195) and have added a new figure (Figure 10) showing the recommended iteration and added a reference to a supplemental file, which includes the recommended project brief (lines 538-545; Supplemental File #1) that will be used in future offerings of this project based on lessons learned from this investigation.

8. Please, in conclusions explain future works, or is this a close way in teaching? I think not, so an explanation is needed.

We have revised the conclusions to better address this future work. Due to the mixed results in the third iteration, a fourth iteration has been developed and is now presented, which will be tested in the upcoming academic year. Figure 10 has been provided to visualize this recommended process, and a supplemental file has been provided containing a synthesis of the Iteration 2 and Iteration 3 design briefs to permit implementation of this project by others. It is hoped that should other faculty choose to implement this project, their findings be shared with the author so that the applicability of this project in other contexts can be evaluated. This discussion has been added as new content included in lines 535-550.

Once again, we would like to thank the reviewer for their time and care in providing their feedback, which we have found to be both comprehensive and valuable, and which has helped us to substantially improve the clarity and pedagogical value of this paper.

Reviewer 2 Report

In my opinion, this is an article that is not suitable for the Buildings magazine. It is a demonstration of experience in the bachelor's teaching process using BIM methods. The authors describe four basic pedagogical theories applied in BIM teaching. Content seeks to apply BIM and evaluate the energy efficiency of buildings. In some case studies and their modifications, they point to the need for energy (heating, cooling) without further justification. The benefits are in education. It would be useful for authors to consider placing an article in a journal that is dedicated to teaching science. The article has no novelty. It only shows the experience and results of the pedagogical process without thinking about the consequences. I do not recommend publishing in a form such as, but completely reworking. The article is also unsatisfactory with its graphical outputs. The figures are unclear and, in particular, the description is unreadable (Figs. 4,6,7,9,11). Where are the pictures 8,10 ???
The conclusions are well known and merely state the use of BIM in teaching process (lines 443-450).

Author Response

We thank the reviewer for taking their time to review this manuscript and have taken their comments as a whole as an indication of the need for additional clarity regarding the framing of the manuscript and its topic. Our detailed responses to each comment are shown in blue text below.

In my opinion, this is an article that is not suitable for the Buildings magazine. It is a demonstration of experience in the bachelor's teaching process using BIM methods. The authors describe four basic pedagogical theories applied in BIM teaching. Content seeks to apply BIM and evaluate the energy efficiency of buildings. In some case studies and their modifications, they point to the need for energy (heating, cooling) without further justification. The benefits are in education. It would be useful for authors to consider placing an article in a journal that is dedicated to teaching science.

While we appreciate the reviewer’s suggestion that this instead be submitted to a pedagogical journal (and agree that it would be a very strong fit in such a context), we note that this was not submitted to the Buildings journal per se, but was explicitly submitted to the IT in Construction special issue for this journal, consisting of papers selected from the CIB W78 2018 conference that were specifically invited by the scientific committee based on their merit. We respectfully request that the question of fit be deferred to the guest editor of this special issue who requested that this paper be submitted in full-length format. As regards the case studies, these are presented to show how students engaged with the energy modeling results to identify high relative envelope losses or improved performance associated with different material and massing selections and the focus of the paper was intended to show how students engaged with these results to improve the energy performance of their design, rather than discuss the details of the heating and cooling loads for each, as the latter is not within the paper’s scope.

The article has no novelty. It only shows the experience and results of the pedagogical process without thinking about the consequences. I do not recommend publishing in a form such as, but completely reworking.

We respectfully disagree. This paper has significant novelty, as no other academic literature has been found that discusses a project – let alone a four year study – to apply data-driven design as a means of developing advanced building information modeling skills. A recent paper (Coates et al, 2018) highlighted the need to lead students to “consider future potentials” and noted the need to “be able to apply analysis techniques to BIM models “ but no examples of a completed study of such an implementation was noted.  We have modified the introductory text (lines 55-61) to better frame this context in response to this comment. Should the reviewer know of other published studies showing how data-driven design has been used for teaching advanced BIM skills, we would welcome that information as a comparison of conclusions of such studies would be extremely helpful for the corresponding author in future offerings of this project and we have been unable to find a single instance of such a case for comparison.

Coates, S.P., Biscaya, S. and Rachid, A., 2018. The utilization of BIM to achieve prescribed undergraduate learning outcomes. Available online:  http://usir.salford.ac.uk/id/eprint/48318/1/THE%20UTILIZATION%20OF%20BIM%20TO%20ACHIEVE%20PRESCRIBED%20ARCHITECTURAL%20UNDERGRADUATE%20LEARNING%20OUTCOMES.pdf

The article is also unsatisfactory with its graphical outputs. The figures are unclear and, in particular, the description is unreadable (Figs. 4,6,7,9,11). Where are the pictures 8,10 ???

The descriptions of the case studies cannot be shown at the scale of image supported by this journal and will be transcribed into a supplemental file (Supplemental File #2) developed to accompany this paper.

The conclusions are well known and merely state the use of BIM in teaching process (lines 443-450).

While we agree that these are known outcomes of effective BIM teaching, the intention of noting these outcomes in the conclusion were to show precisely this: that this novel approach to BIM teaching has been extremely effective in achieving the desired outcome of BIM.

That said, since this is a Buildings focused journal, we refer the reviewer to the discussion of sustainable design learning insights gained by the students in lines 503-509 and note the significance of this experiential learning in the context of the recently released ASHRAE 209-2018 standard. Further, we highlight the additional section (Section 5) showing a more advanced application of the project concepts completed by a student after completing the project to demonstrate how the knowledge gained has permitted the design of a net-zero building using iterative data-driven design within a BIM context.

Again, we thank this reviewer for taking the time to review this paper.

Round 2

Reviewer 2 Report

I appreciate the authors' opinion that the article was not submitted to the Buildings magazine, but was explicitly submitted to a special issue....

...we note that this was not submitted to the Buildings journal per se, but was explicitly submitted to the IT in Construction special issue for this journal, consisting of papers selected from the CIB W78 2018 conference that were specifically invited by the scientific committee based on their merit. We respectfully request that the question of fit be deferred to the guest editor of this special issue who requested that this paper be submitted in full-length format.

I appreciate the suggestion that the authors ask respectfully that the question of suitability be postponed to the guest editor of this special edition who requested that this document be submitted in full format.

I agree, but in my opinion (does not affect the guest editor's opinion) the article does not fit into the Buildings magazine. I stand on my point of view.

For case studies, these are presented to show how energy modeling students identify with high relative envelope losses or better performance associated with different material and mass selections, and the focus of the paper was to show how students working with these results. This fact again highlights the fact that it is a pedagogical process and not professional results concerning the construction of buildings.

I appreciate the authors' opinion that they do not respect my opinion with respect. I respectfully disagree that no other academic literature is known and this work has a major innovation.

The authors write: The descriptions of the case studies cannot be shown at the scale of image supported by this journal and will be transcribed into a supplemental file (Supplemental File #2) developed to accompany this paper.

I think that even after reworking, the article is unsatisfactory with its graphical outputs. The figures are unclear and, in particular, the description is unreadable.

Yes Supplemental File # 2 - Full Transcript of Figure Text
It has unreadable texts explained for figure 3, 4, 5, 6, and 8, 9. This is a complex description on 8 pages at the expense of picture readability. Then why are these unreadable images in the text? Looks like if the authors took some student posters and made them smaller, enough to be accepted in this article. Would I suggest putting these images in a readable form and not solving it with an additional file.

The authors write: While we agree that these are known outcomes of effective BIM teaching, the intention of noting these outcomes in the conclusion were to show precisely this: that this novel approach to BIM teaching has been extremely effective in achieving the desired outcome of BIM.

This fact confirms my opinion that the article is suitable for another journal and not for the Buildings magazine.